# Case Series of Patients with Coronavirus Disease 2019 Pneumonia Treated with Hydroxychloroquine

**DOI:** 10.3390/medicina59030541

**Published:** 2023-03-10

**Authors:** Tomohiro Tanaka, Masaki Okamoto, Norikazu Matsuo, Yoshiko Naitou-Nishida, Takashi Nouno, Takashi Kojima, Yuuya Nishii, Yoshihiro Uchiyashiki, Hiroaki Takeoka, Yoji Nagasaki

**Affiliations:** 1Department of Respirology and Clinical Research Center, National Hospital Organization Kyushu Medical Center, 1-8-1 Jigyohama, Chuo-ku, Fukuoka 810-0065, Japan; 2Division of Respirology, Neurology and Rheumatology, Department of Internal Medicine, Kurume University School of Medicine, 67 Asahi-machi, Kurume, Fukuoka 830-0011, Japan; 3Department of Infectious Disease, National Hospital Organization Kyushu Medical Center, 1-8-1 Jigyohama, Chuo-ku, Fukuoka 810-0065, Japan

**Keywords:** coronavirus disease 2019, hydroxychloroquine, severe acute respiratory syndrome

## Abstract

The efficacy of hydroxychloroquine (HCQ) therapy, a previous candidate drug for coronavirus disease 2019 (COVID-19), was denied in the global guideline. The risk of severe cardiac events associated with HCQ was inconsistent in previous reports. In the present case series, we show the tolerability of HCQ therapy in patients treated in our hospital, and discuss the advantages and disadvantages of HCQ therapy for patients with COVID-19. A representative case was a 66-year-old woman who had become infected with severe acute respiratory syndrome coronavirus 2 and was diagnosed as having COVID-19 pneumonia via polymerase chain reaction. She was refractory to treatment with levofloxacin, lopinavir, and ritonavir, while her condition improved after beginning HCQ therapy without severe side effects. We show the tolerability of HCQ therapy for 27 patients treated in our hospital. In total, 21 adverse events occurred in 20 (74%) patients, namely, diarrhea in 11 (41%) patients, and elevated levels of both aspartate aminotransferase and alanine transaminase in 10 (37%) patients. All seven grade ≥ 4 adverse events were associated with the deterioration in COVID-19 status. No patients discontinued HCQ treatment because of HCQ-related adverse events. Two patients (7%) died of COVID-19 pneumonia. In conclusion, HCQ therapy that had been performed for COVID-19 was well-tolerated in our case series.

## 1. Introduction

The novel pathogen severe acute respiratory syndrome coronavirus 2 (SARS-CoV-2) caused an outbreak of viral pneumonia that became known as coronavirus disease 2019 (COVID-19), and was first reported in Wuhan, China in December 2019 [1]. After the initial outbreak, the illness rapidly spread globally [1].

The clinical studies of several drug candidates for COVID-19 therapy were globally performed [2,3]. However, the only drugs with evidence supporting reduced mortality against COVID-19 are corticosteroids. The RECOVERY trial, a controlled, open-label trial, suggested that 28-day mortality was lower in patients with moderate or severe COVID-19 who had received dexamethasone compared with those who had received standard care alone [4]. However, no benefit was seen in patients with mild COVID-19 [4]. In a prospective meta-analysis of 10,930 patients with COVID-19, compared with standard care or placebo, the administration of tocilizumab and interleukin-6 antagonists was associated with lower 28-day all-cause mortality [5].

Hydroxychloroquine (HCQ) and the 15-member macrolide antibiotic azithromycin (AZM) have been used for COVID-19 treatment, but were denied in therapeutic guidelines [6,7,8,9,10,11,12]. HCQ and chloroquine phosphate, which are widely used antimalarial drugs that are also used to treat autoimmune diseases such as systemic lupus erythematosus, exert antiviral effects by increasing the endosomal pH to a level exceeding that required for viral/cell fusion [13]. These drugs also interfere with the glycosylation of cellular receptors for SARS-CoV-2 [6,7,8,9,13]. Because macrolides prevent the production of proinflammatory mediators, cytokines, and reactive oxygen species both in vitro and in vivo [14], and control the exacerbation of underlying respiratory diseases such as asthma, panbronchiolitis, acute respiratory distress syndrome, and chronic fibrosing interstitial pneumonia [15,16,17], a combination therapy with HCQ and AZM has been used to treat COVID-19. SARS-CoV-2 real-time polymerase chain reaction-negative conversion rates in patients with COVID-19 after combined treatment with HCQ and AZM were 83% and 93% on disease Days 7 and 8, respectively [9]. However, a randomized, controlled, open-label platform (RECOVERY) trial compared the 28-day-mortality rates of 1561 patients who had received hydroxychloroquine and 3155 patients who had received standard care; the rate did not differ between the groups [10]. Additionally, retrospective cohort studies revealed that treatment with HCQ and/or AZM was not associated with significant differences in the incidence of intubation or death or the rate of inhospital mortality [11,12]. A recent WHO guideline recommended against administering HCQ for treatment of COVID-19 patients [18]. Moreover, there are concerns that HCQ may cause cardiovascular events, such as arrhythmia or cardiac arrest, in patients with COVID-19. Chorin et al. observed that, among 80 hospitalized SARS-CoV-2-infected patients who had received HCQ plus AZM, 30% exhibited corrected QT interval (QTc) prolongation of >40 ms, and 11% exhibited QTc prolongation of >500 ms [19]. The increase in the risk in cardiac arrest by HCQ and/or AZM is caused from drug-induced QT-interval prolongation and torsades de pointes (a form of polymorphic ventricular tachycardia) [19,20]. A retrospective study performed in New York suggested that the incidence of cardiac arrest was significantly higher in patients receiving both HCQ and AZM (15.5%), but not in those receiving HCQ (13.7%) or AZM (6.2%) alone, compared with patients who had received neither drug (6.8%) [12]. There were also no significant differences in the relative likelihood of abnormal electrocardiographic findings between the patient groups [12]. In the RECOVERY trial, patients who had received HCQ had a greater risk of death from cardiac causes (mean excess, 0.4%), although there was no difference in the incidence of new major cardiac arrhythmia compared with that in patients who had received standard care [10]. However, it is not yet clear whether treatment with HCQ and/or AZM increases the risk of cardiac events among Japanese patients with COVID-19.

In the present case series, we show the tolerability of HCQ therapy in patients treated in our hospital, and we discuss the advantages and disadvantages of HCQ therapy that had been performed for COVID-19 patients.

## 2. Case Report

### 2.1. Representative Patient

The clinical course and chest high-resolution computed tomography (HRCT) images of a representative case are shown in Figure 1. A 66-year-old woman visited her local hospital with intermittent shivering and a fever of >38 °C that improved without intervention in February 2020. Three days later, she visited our hospital in compliance with directions from the health center on the suspicion of a COVID-19 infection contracted from her husband, who had been diagnosed with COVID-19 pneumonia. The patient had no symptoms, and her chest radiographs were normal. However, patchy peripheral ground-glass opacities involving the subpleural area were visible in the lower-right lung lobe that were consistent with previously reported findings of COVID-19 pneumonia. She was admitted to our hospital and was diagnosed with COVID-19 pneumonia on the basis of positive results from a polymerase chain reaction (PCR) assay for SARS-CoV-2 from an oropharyngeal swab sample. Test results from similar samples for antigens of influenza, respiratory syncytial virus, and adenovirus were negative. The patient’s medical history included only a mackerel allergy, and she had no history of smoking. Her vital signs on admission were as follows: respiratory rate, 18 breaths/min; oxygen saturation (SpO_2_), 96% (room air); heart rate, 96 beats/min; blood pressure, 151/70 mmHg; and body temperature, 36.7 °C. Auscultation revealed no abnormal respiratory or heart sounds. Hematology and other laboratory examinations showed slightly elevated C-reactive protein (CRP; 0.62 mg/dL) and lactate dehydrogenase (LDH; 245 IU/mL) levels, with lymphocytopenia (610 cells/µL). The patient’s clinical course and chest HRCT images are shown in Figure 1. At hospital admission, the patient was asymptomatic. However, on Day 4 after admission, she developed a fever (38.3 °C) and began treatment with levofloxacin (500 mg/day), lopinavir (800 mg/day), and ritonavir (200 mg/day). On Day 6, the lopinavir and ritonavir were discontinued because the patient developed diarrhea that was suspected to be an associated adverse effect. On Day 7, her condition deteriorated: her body temperature had increased to 39.1 °C, with elevated CRP (8.25 mg/dL) and LDH levels (353 IU/L), and decreased SpO_2_ (91% on room air) compared with previous values. Therefore, we started treatment with hydroxychloroquine (400 mg/day). On Day 7, HRCT showed newly appeared ground-glass opacities in the right lung lobe. Interlobular septal thickening, perilobular opacities, and curvilinear lines were also observed in the peripheries of both lungs. After treatment with hydroxychloroquine, the patient’s fever improved on Day 9, and all other symptoms improved on Day 15, as follows: CRP, 0.19 mg/dL; LDH, 208 IU/L; and SpO_2_, 98% on room air. We confirmed that results from a PCR assay for SARS-CoV-2 were negative, and we stopped treatment with hydroxychloroquine. The patient was discharged on Day 16, and follow-up HRCT images obtained on Day 19 showed improvement in the lung changes. Her condition was stable after discharge. Grade 1 elevations in aspartate aminotransferase (AST) and alanine aminotransferase (ALT) levels, in accordance with the Common Terminology Criteria for Adverse Events version 5.0, both of which increased on Day 11 and normalized on Day 26, were suspected to be adverse effects of HCQ therapy.

### 2.2. Outcome of COVID-19 Patients Treated with HCQ

The patient characteristics, outcomes, and safety and tolerability of therapy in patients with COVID-19 who had received HCQ in our hospital in 2020 are shown in Table 1 and Table 2. A total of 27 patients (23 males; median age, 56.0 years), including Cases 1 and 2, were diagnosed with COVID-19 pneumonia via SARS-CoV-2 PCR and were treated with HCQ. COVID-19 severity in the 27 patients was mild in 6 (22%), moderate in 12 (44%), and severe in 9 (33%). Of the 27 patients, 18 (66%) received concurrent AZM, 8 (30%) received a concurrent corticosteroid, 3 (11%) received concurrent favipiravir, 2 (7%) received concurrent lopinavir and ritonavir, 2 (7%) received concurrent tocilizumab, and 1 (4%) received concurrent remdesivir. Of the patients, 21 required oxygen therapy, namely, mechanical ventilation for 4 patients, mask with a reservoir for 5, and nasal cannula for 12. The remaining 6 patients did not require oxygen therapy.

### 2.3. Tolerability of Treatment with HCQ in Patients with COVID-19

Table 2 shows the administration dose and duration of HCQ and AZM therapy in previous studies and the present study. In the present study, 400 mg of HCQ was administered to all patients, and the median duration of treatment was 10.0 (6.0–12.0) days. In 14 of the 18 patients treated with AZM, 500 mg was administered intravenously for 3.0 (3.0–3.5) days, and 4 patients received 2000 mg for 1 day. In a retrospective study in New York, 908 (90.3%) of 1006 HCQ-treated patients received 400 mg of AZM [12]. Among 946 AZM-treated patients, 870 (92.0%) received a dose of 500 mg [12]. Similarly, AZM was administered orally to 454 (48.0%) patients and intravenously to 482 (50.9%) patients [12]. Although the administration period of HCQ was not stated, the median hospital stay was 7 days in both the HCQ + AZM and HCQ monotherapy groups in the study [12]. In a cohort study in France, HCQ was administered at a dose of 600 mg daily for 10 days combined with AZM at a dose of 500 mg on Day 1, followed by 250 mg daily for the next 4 days [9]. In a cohort study in New York City, HCQ was administered at a dose of 600 mg on Day 1 followed by 400 mg daily for a median of 5 days [11]. In the RECOVERY trial, 800 mg of HCQ was administered at 0 and 6 h followed by a maintenance dose of 400 mg every 12 h for a median of 6 days (interquartile range, 3–10 days) [10]. In our study, there was no significant difference in the cumulative dose and duration of treatment with HCQ and AZM compared with previous reports.

The details of the adverse events (AEs) that had occurred during HCQ treatment are presented in Table 3. We observed 47 AEs in 20 (74%) of 27 patients with COVID-19. The AE severities in accordance with the Common Terminology Criteria for Adverse Events version 5.0 were Grade 1 in 5 (19%) patients, Grade 2 in 8 (30%) patients, Grade 3 in 1 (4%) patient, Grade 4 in 5 (19%) patients, and Grade 5 in 2 (7%) patients. All Grade ≥ 4 AEs were associated with deterioration of COVID-19. Two patients eventually died of COVID-19, with the deaths occurring 44 days after the start of HCQ treatment in one patient, and 18 days after admission in the second patient. The most frequent AE was diarrhea, occurring in 11 of 27 (41%) patients. The most frequent AE in the blood laboratory tests was elevated AST and ALT levels, which occurred in 10 of 27 (37%) patients. Neither arrhythmia nor cardiac arrest was observed in any patient. Bacterial pneumonia occurred after beginning HCQ therapy in two patients who had been diagnosed with ventilation-associated pneumonia caused by methicillin-resistant *Staphylococcus aureus*. One of these patients developed sepsis, and methicillin-resistant *Staphylococcus aureus* was detected via blood culture. Of the 11 patients with diarrhea during hospitalization in the infectious disease ward, 9 (82%) recovered. The duration from the initial appearance of diarrhea to recovery was 1.0 (1.0–4.5) days. In the patients with AST and ALT elevation, the elevations resolved in 6 (60%) patients, and the durations from the appearance of AST and ALT elevation to recovery were 14.0 (9.0–35.0) and 9.5 (3.8–23.8) days, respectively. The discontinuation of HCQ was required in only two patients, both of whom died of deterioration of COVID-19. No patient discontinued HCQ treatment because of HCQ-related AEs. Furthermore, no patients required an HCQ dose reduction. Our data indicate no severe HCQ-related AEs that required the discontinuation of the drug.

## 3. Discussion

We reported a clinical case series of patients who had received HCQ for COVID-19. As the data in patients treated with HCQ therapy were not compared with those in the control group, it is not certain that the adverse events were caused in the present case series by COVID-19 or HCQ therapy. One of the most frequent AEs in the present study was diarrhea. This result is similar to the findings in previous studies of chloroquine or HCQ treatment that reported that gastrointestinal involvement, typically of mild severity, is one of the main AEs associated with this regimen [21,22]. Conversely, two clinical studies on COVID-19 reported diarrhea in only 4 of 80 (5%) patients and 85 of 735 (11.6%) patients who had been treated with HCQ [9,12]. Diarrhea may be caused by gastrointestinal tract infection in patients with COVID-19. A previous cohort study found that 2–10% of patients with COVID-19 presented with diarrhea, and SARS-CoV-2 RNA was detected in both stool and blood samples [23]. The second most frequent AE in this study was elevated AST and ALT levels. However, this was an uncommon finding in previous reports of HCQ therapy; thus, the main cause of transaminase elevation may not have been HCQ therapy [19,21]. Previous cohort studies of patients with COVID-19 in China indicated that the incidence of transaminase elevation ranged from 16.1% to 53.1% [1,24,25]. Transaminase elevation in patients with COVID-19 may be caused by direct liver injury by SARS-CoV-2 via angiotensin-converting enzyme-2 in cholangiocytes, cytokine storm, or pneumonia-associated hypoxia [24,25]. In the present study, all episodes of diarrhea and transaminase elevation were Grade 1–2 in severity, and none required HCQ discontinuation.

In contrast to previous studies on HCQ therapy for COVID-19, none of our patients exhibited arrhythmia or cardiac arrest after commencing HCQ and/or AZM therapy [12]. Electrocardiograms performed in a previous study of patients treated with HCQ for connective tissue disease revealed that the heart conditions of these patients, including QTc intervals, were not different from those of healthy controls [26]. The rate of heart conduction disorders was similar to that expected in the general population. The cause of the difference in the incidence of arrhythmia and cardiac arrest between previous studies and the present study is unknown. Twelve-lead electrocardiogram (ECG) examination on admission or ECG monitoring were not performed in all of the present case series. One of the study limitations is that diagnostic tools for detecting arrhythmia may differ from those of previous studies of HCQ and/or AZM therapy. An important risk factor for arrhythmia associated with HCQ exposure is QTc prolongation. The administration of certain drugs, such as H2 blockers and antipsychotics, can cause long QT syndrome (LQTS) [20]. Additionally, race-related differences in the prevalence of LQTS-related gene mutations could have influenced our study results. The compound mutations of LQTS-related genes were observed in 8.4% of 310 Japanese probands with genotyped LQTS [27]. Moreover, Itoh et al. reported that the gene mutation causing congenital LQTS was present in patients diagnosed with drug-induced LQTS [28]. We could not perform genetic testing in the present study. This limitation of the present study should be addressed in future research. The cumulative doses and durations of treatment with HCQ and AZM in the present study were not lower than those of previous observational studies and randomized controlled trials; therefore, it is unlikely that any differences affected drug tolerability [10,15,16,17]. The cause of the difference in the incidence of severe cardiac events between previous studies and the present study may be related to other factors, such as differences in the incidence of QT prolongation associated with race or concomitant drug therapy. The present case series supports the tolerability of HCQ therapy in Japanese patients with COVID-19. Recently, Samuel et al. discussed that HCQ may have potential as a therapeutic agent for long COVID-19 but not acute symptoms because HCQ can inhibit unremitting inflammatory response, MHC class II-mediated autoantigen presentation, a sustained endotheliopathy due to microthrombi shown in long COVID-19. However, more prospective trials are needed to prove the therapeutic efficacy of HCQ for long COVID-19 [29,30].

The present case series had some limitations. First, the present case series had a small population and were not compared with control subjects. Second, as mentioned above, the diagnostic tools for detecting arrhythmia were not unified. However, the preliminary results of tolerability of HCQ therapy for Japanese COVID-19 patients may contribute to the study of HCQ for other conditions such as long COVID-19.

## 4. Conclusions

In conclusion, treatment with HCQ for COVID-19 was well-tolerated in our case series.

## Figures and Tables

**Figure 1 medicina-59-00541-f001:**
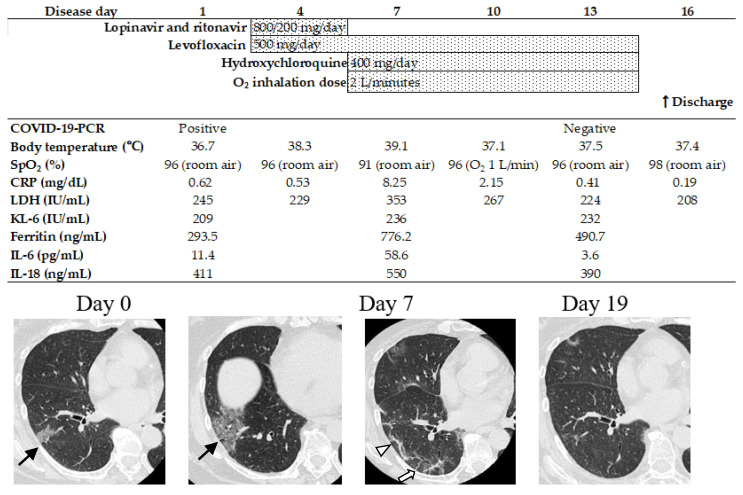
Clinical course of a representative case. O_2_, oxygen; COVID-19, coronavirus disease 2019; PCR, polymerase chain reaction; SpO_2_, oxygen saturation; CRP, C-reactive protein; LDH, lactate dehydrogenase; KL-6, Krebs von den Lungen-6, ground-glass opacitie; black arrow, interlobular septal thickening; white arrow, perilobular opacities and curvilinear line; arrow head.

**Table 1 medicina-59-00541-t001:** Patient characteristics.

N	27	Pharmacological therapy	
Age (years)	56.0 (46.0–72.0)	HCQ alone	27 (100%)
Gender	23 (85%)	HCQ with AZM	18 (66%)
Smoker	12 (44%)	Favipiravir	3 (11%)
e: mild/moderate/severe	6 (22%)/12 (44%)/9 (33%)	Lopinavir and ritonavir	2 (7%)
Data at admission		Remdesivir	1 (4%)
Count of blood cells		Corticosteroid	8 (30%)
White blood cells (/uL)	4500.0 (3600.0–6000.0)	Tocilizumab	2 (7%)
Neutrophils (%)	73.2 (65.6–76.8)	Oxygen therapy	
Lymphocytes (%)	16.9 (14.0–25.2)	Nasal cannula	12
Platelets (×10^4^/uL)	17.8 (14.1–21.4)	Oxygen mask with reservoir	6
Laboratory data		Mechanical ventilation	3
CRP (mg/dL)	4.6 (1.4–9.3)		
Lactate dehydrogenase (IU/L)	320.0 (234.0–471.0)		
Ferritin (ng/mL)	781.1 (371.8–1222.7)		
Interleukin-6 (pg/mL)	16.9 (11.6–58.3)		
D-dimmer (ug/mL)	0.90 (0.50–1.2)		

HCQ, hydroxychloroquine; AZM, azithromycin; SARS-CoV2, Severe acute respiratory syndrome coronavirus; PCR. polymerase chain reaction.

**Table 2 medicina-59-00541-t002:** Present and previous reports of HCQ therapy.

			Administration Dose (mg/day)		Duration of Administration (Days)		Main
	Design	N	HCQ	AZM	HCQ	AZM	Side Effect
Horby et al. [10]	RCT	4716	800 (loading dose) †		6.0		Risk of death from
(RECOVERY trial)			400 (maintenance dose)				cardiac causes
Geleris et al. [11]	Observational	1376	600 (loading dose)		5.0		N.D.
	study		400 (maintenance dose)				
Rosenberg et al. [12]	Retrospective	1438	400 in	500 in	N.D.	N.D.	Cardiac arrest in
	study		90.3% of subjects	92.0% of subjects			HCQ and AZM group
Gautret et al. [9]	Retrospective	80	600	500 (loading dose)	10.0	5.0	N.D.
	study			250 (maintenance dose)			
The present study	Retrospective	27	400	500 (p.o.)	10.0 (6.0–12.0)	3.0 (3.0–3.5) (p.o.)	Diarrhea
	study			or 2000 mg (i.v.)		1.0 (i.v.)	

HCQ, hydroxychloroquine; AZM, azithromycin, RCT, randomized controlled trial; p.o., per oral; i.v., intravenous; N.D., no data. † 800 mg of HCQ were administered at 0 and 6 h followed by a maintenance dose of 400 mg every 12 h.

**Table 3 medicina-59-00541-t003:** Adverse events experienced by patients.

		Recovering	Duration (days)		Grade of CTCAE ver. 5.0				
		Yes/No	From Start of HCQ to Appearance of AE	From Appearance to Improvement of AE	1	2	3	4	5
Number	27								
Incidence of more than 1 AEe	20 (74%)								
Highest severity of AE evaluated					5 (19%)	8 (30%)	1 (4%)	5 (19%)	2 (7%)
by CTCAE ver. 5.0-grade									
Discontinuation of HCQ	2 (7%)								
Cause	Death of COVID-19 2								
Adverse event									
Diarrhea	11 (41%)	9/2	1.0 (1.0–4.0)	1.0 (1.0–4.5)	11	0	0	0	0
Appetite loss	1 (4%)	1/0	1.0	2.0	0	1	0	0	0
Cough	2 (7%)	1/1	4.5 (4.0–5.0)	3.0	2	0	0	0	0
Elevation of aspartate transaminase	10 (37%)	6/4	4.0 (3.0–8.8)	14.0 (9.0–35.0)	6	2	2	0	0
Elevation of alanine transaminase	10 (37%)	6/4	4.5 (3.0–9.5)	9.5 (3.8–23.8)	6	1	3	0	0
Elevation of creatinine	1 (4%)	1/0	13.0	38.0	0	0	1	0	0
Thrombocytopenia	1 (4%)	0/1	29.0		0	0	1	0	0
Bacterial pneumonia	2 (7%)	2/0	7.0 (3.0–11.0)	13.5 (9.0–18.0)	0	0	2	0	0
Sepsis	1 (4%)	0/1	42.0		0	0	1	0	0
Guillain–Barre syndrome	1 (4%)	1/0	14.0	11.0	0	1	0	0	0
Deterioration of COVID-19	7 (26%)	5/2	2.0 (1.0–8.0)	13.0 (5.5–15.0)	0	0	0	5	2

AE, adverse event; CTCAE, Common Terminology Criteria for Adverse Events; HCQ, hydroxychloroquine; COVID-19, coronavirus disease-2019.

## Data Availability

The data presented in this study are available on request from the corresponding author. The data are not publicly available due to ethical considerations.

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
