# Peer review of "Case Series of Patients with Coronavirus Disease 2019 Pneumonia Treated with Hydroxychloroquine"

_medicina, 2023, doi:10.3390/medicina59030541_

Round 1
Reviewer 1 Report
Authors reported the outcomes from COVID-19 patients with pneumonia (n=27) receiving HCQ therapy through a case series. The most highlighted outcome in the paper is the adverse effect of HCQ – diarrhea. Increased level of serum AST and ALT levels were also reported. Interestingly, cardiac events were not reported in this study. Please find below my comments:
1. Please provide the updates on anti-SARS-CoV-2 therapies in the introduction. Author may refer to these studies:
Sharun et al. 2022. Narra J. 2(3): e92
Masyeni et al. 2022. Journal of Medical Virology. 94(7): 3006-3016
2. Authors argued that the adverse events were likely to be caused by the COVID-19 itself, not the therapy. The absence of control group in this study makes it difficult to reach such conclusion.
Please emphasize the long COVID-19 symptoms (Fahriani et al. Narra J. 1(2): e36) and their association with HCQ intake (Wang and Xu. Clinical Rheumatology (2023) doi: 10.1007/s10067-023-06514-x).
3. Apparently, it is not clear how HCQ could cause arrhythmia or cardiac arrest. Supposedly, there are explanations from the reported studies about the probable cause of HCQ-induced arrhythmia or cardiac arrest – please explain. Moreover, could you also make a comparison between the diagnostic tools used in the reported studies and this present study; probably the different finding about cardiac events is caused by this difference(?)
4. Please provide a specific sub-heading for the outcomes of this therapy in the results.
5. Please make an additional last paragraph in ‘discussion’ explaining about the strengths and limitations of the reported study. Moreover, please also emphasize the what your study could offer to the drug development or the faith of HCQ.
Author Response
Reviewer 1
Comment 1. Please provide the updates on anti-SARS-CoV-2 therapies in the introduction. Author may refer to these studies:
Sharun et al. 2022. Narra J. 2(3): e92
Masyeni et al. 2022. Journal of Medical Virology. 94(7): 3006-3016
Response
Thank you for your comment. I have added these articles to reference section (Reference No. 2 and 3).
Comment 2-1. Authors argued that the adverse events were likely to be caused by the COVID-19 itself, not the therapy. The absence of control group in this study makes it difficult to reach such conclusion.
Response
As the reviewer showed, no evidence that adverse events were caused by COVID-19 itself, not therapy. I have revised the description as follows.
Line 207, Discussion
As the data in patients treated with HCQ therapy were not compared with those in control group, it is not exactly that the adverse events are caused shown in the present case series by COVID-19 or HCQ therapy.
Comment 2-2. Please emphasize the long COVID-19 symptoms (Fahriani et al. Narra J. 1(2): e36) and their association with HCQ intake (Wang and Xu. Clinical Rheumatology (2023) doi: 10.1007/s10067-023-06514-x).
Response
We have added articles shown by the reviewer to reference section (Reference No. 29 and 30) and discussed the HCQ therapy and symptom caused from long COVID-19.
Line 253, Discussion
Recently, Samuel et al discussed that HCQ may have a potential as therapeutic agent for long COVID but not acute symptom, because HCQ can inhibit unremitting inflammatory response, MHC class II-mediated autoantigen presentation, a sustained endotheliopathy due to microthrombi shown in long COVID-19. However, the further prospective trials are needed to prove the therapeutic efficacy of HCQ for long COVID-19 [29-30].
Comment 3
- Apparently, it is not clear how HCQ could cause arrhythmia or cardiac arrest. Supposedly, there are explanations from the reported studies about the probable cause of HCQ-induced arrhythmia or cardiac arrest – please explain. Moreover, could you also make a comparison between the diagnostic tools used in the reported studies and this present study; probably the different finding about cardiac events is caused by this difference(?)
Response
Thank you for your comment. We explained how HCQ could cause arrhythmia or cardiac arrest in discussion section.
As reviewer say, the difference of diagnostic tool in between the present and previous study. We added the study limitation in Discussion section as follows.
Line 71, Introduction
The increase of the risk in cardiac arrest by HCQ and/or AZM is caused from drug-induced QT-interval prolongation and torsades de pointes (a form of polymorphic ventricular tachycardia) [19, 26].
Line 235, Discussion
Twelve-lead Electrocardiogram (ECG) examination on admission or ECG monitoring were not performed in all of the present case series. One of the study limitations is that diagnostic tools for detecting arrhythmia may differ from previous studies of HCQ and/or AZM therapy.
Comment 4
- Please provide a specific sub-heading for the outcomes of this therapy in the results.
Response
Thank you for your comment. We divided the clinical course after HCQ therapy for 27 cases (2.2 Tolerability of Treatment with HCQ in Patients with COVID-19) into two sections on outcome and tolerability.
Comment 5
- Please make an additional last paragraph in ‘discussion’ explaining about the strengths and limitations of the reported study. Moreover, please also emphasize the what your study could offer to the drug development or the faith of HCQ.
Response
 Thank you for your comment. We added limitation, strength, and possible contribution for drug development in this case series in discussion section as follow.
Line 259, Discussion
The present case series had some limitations. First, the present case series have small population and were not compared with control subject. Second, as mentioned above, the diagnostic tools for detecting arrhythmia were not unified. However, the preliminary results of tolerability of HCQ therapy for Japanese COVID-19 patients may contribute to study of HCQ for other conditions such as long COVID-19.
Reviewer 2 Report
The authors in there manuscript 'Case Series of Patients with Coronavirus Disease 2019 Pneumonia Treated with Hydroxychloroquine' have done case reporting to show that treatment with HCQ for COVID-19 was well-tolerated. The result of the case study does suggest the same.
However, WHO does not recommend use of hydroxychloroquine as a treatment for COVID-19. Because use of hydroxychloroquine did not reduce mortality or the need for or duration of mechanical ventilation. The recommendation of WHO is based on many clinical trials and the details may be found on the WHO website.
In light of WHO recommendation it is important to mention/discuss WHO recommendation in the Introduction/Discussion section of the manuscript to clarify the objective of this case study. Also the authors need to add their observation/recommendation that even if HCQ is well tolerated in COVID-19 patient whether there is any benefit or need for the use of HCQ in future for the treatment and management of COVID-19 patients.
Author Response
Reviewer 2
Comment
In light of WHO recommendation it is important to mention/discuss WHO recommendation in the Introduction/Discussion section of the manuscript to clarify the objective of this case study. Also the authors need to add their observation/recommendation that even if HCQ is well tolerated in COVID-19 patient whether there is any benefit or need for the use of HCQ in future for the treatment and management of COVID-19 patients.
Response
Thank you for your important comment.
I agree with reviewer comment. We added the following sentences in Discussion section for clarifying that HCQ therapy is not recommended for COVID-19 and added the WHO guideline as reference (reference No. 13).
 Moreover, we have added articles shown by the reviewer to reference section (Reference No. 29 and 30) and discussed the HCQ therapy and symptom caused from long COVID-19. Please show the response to comment 2-2 from reviewer 1.
Line 66, Introduction
Recently, WHO guideline recommends against administering HCQ for treatment of COVID-19 patients [13].
Line 253, Discussion
Recently, Samuel et al discussed that HCQ may have a potential as therapeutic agent for long COVID but not acute symptom, because HCQ can inhibit unremitting inflammatory response, MHC class II-mediated autoantigen presentation, a sustained endotheliopathy due to microthrombi shown in long COVID-19. However, the further prospective trials are needed to prove the therapeutic efficacy of HCQ for long COVID-19 [29-30].
HCQ is safe and widely used in rheumatology for patients with SLE, and rheumatoid arthritis, which are prototypes of chronic inflammatory/autoimmune disease.
Round 2
Reviewer 2 Report
No further comments